# Dosage sensitivity is a major determinant of human copy number variant pathogenicity

Alan M. Rice[1] & Aoife McLysaght[1]

Human copy number variants (CNVs) account for genome variation an order of magnitude larger than single-nucleotide polymorphisms. Although much of this variation has no phenotypic consequences, some variants have been associated with disease, in particular neurodevelopmental disorders. Pathogenic CNVs are typically very large and contain multiple genes, and understanding the cause of the pathogenicity remains a major challenge. Here we show that pathogenic CNVs are significantly enriched for genes involved in development and genes that have greater evolutionary copy number conservation across mammals, indicative of functional constraints. Conversely, genes found in benign CNV regions have more variable copy number. These evolutionary constraints are characteristic of genes in pathogenic CNVs and can only be explained by dosage sensitivity of those genes. These results implicate dosage sensitivity of individual genes as a common cause of CNV pathogenicity. These evolutionary metrics suggest a path to identifying disease genes in pathogenic CNVs.

[1] Smurfit Institute of Genetics, Trinity College Dublin, University of Dublin, Dublin 2, Ireland. Correspondence and requests for materials should be addressed to A.McL. (email: aoife.mclysaght@tcd.ie).

Copy number variants (CNVs) are regions of the genome that are duplicated or deleted in some individuals in a population. CNVs are most intensely studied in human but have been observed and characterized to a lesser extent in other species[1–7]. Accounting for more variation than single-nucleotide polymorphisms in terms of base pair length, CNVs are abundant in human genomes. Each individual has on average 1,000 CNVs of >450 bp with respect to the reference genome[8]. CNVs segregate in the population but also arise *de novo*[8,9]. Some regions in the genome are CNV hotspots—~10% of the human genome experiences recurrent CNV events[10].

Often this variation does not produce a phenotype, as CNVs are frequently small, intergenic or encompass genes that can tolerate a change in copy number. Some genes can even be completely deleted with no apparent effect[9]. However, CNVs have previously been associated with a number of human conditions, most notably neurodevelopmental disorders including autism spectrum disorders, schizophrenia, intellectual disability, attention-deficit hyperactivity disorder, developmental delay and epilepsy[11–17]. Owing to this implication in disease, CNVs are subject to increasingly intense scrutiny to understand and characterize their genetic and phenotypic effects.

There is more than one possible mechanism by which a CNV can disrupt gene function and cause a phenotype, including disruption of chromosome structure, interference with regulatory elements and perturbation of relative amounts of dosage-sensitive genes[18]. Several recent studies have shown a relationship between topologically associated domains and genomic duplication effects[19,20]. Still, the prevailing hypothesis on CNV pathogenicity is that it is due to dosage sensitivity of the included genes. One of the first well-characterized cases of CNV pathogenicity was Charcot–Marie–Tooth neuropathy, which was specifically linked to CNV of the dosage-sensitive gene peripheral myelin 22 (*PMP22*)[21]. Dosage sensitivity provides a model whereby a 50% increase or decrease in gene copy number is deleterious[22–25]. Dosage-sensitive genes may be in stochiometric balance with other genes (for example, protein complex members[23]; may operate in a concentration-dependent fashion (for example, developmental morphogens[26] or some splicing co-factors[27]); may produce proteins that are aggregation prone at high concentrations (for example, SNCA[28]); or may have a minimum required concentration to achieve functionality (that is, haploinsufficient genes, including many transcription factors and developmental genes[29]). When the dosage of these genes is changed by an overlapping CNV, the function of the gene is disrupted in a way that we may observe as disease. More acutely dosage-sensitive genes may never be observed in CNVs, even in pathogenic ones, if they are so disruptive as to result in inviability. Thus, duplication and/or loss CNVs of dosage-sensitive genes are not expected to be observed in healthy individuals[30].

As dosage sensitivity is linked to relative abundances rather than absolute amounts, whole-genome duplication (WGD) is tolerable, because, by definition, all genes are duplicated equally. Unlike CNVs and small-scale duplications, WGD events preserve gene stoichiometry. Two such events occurred early in the vertebrate lineage and were followed by extensive genome rearrangement and massive gene loss[31–34]. Duplicated genes retained from these events (ohnologues) are found to be refractory to CNVs and small-scale duplications, that is, they evolve in a pattern that suggests ancient and persistent dosage sensitivity[30]. Ohnologues are depleted among CNVs found in healthy individuals[35] and are found to be overrepresented among genes on pathogenic CNVs[36], providing further supporting evidence that they are under dosage constraint.

Here the evolutionary history of genes in CNVs with different clinical interpretations is examined with the aim of creating a deeper understanding of the predictive power of evolutionary patterns for understanding CNV pathogenicity. We explore the prevailing hypothesis that CNV pathogenicity is frequently due to the copy number change of one or more dosage-sensitive genes or regions found within a variant[37] and predict that this dosage sensitivity will similarly constrain their evolution in mammals in characteristic ways. Consistent with this hypothesis, we find that orthologues of human genes found in pathogenic CNV regions (CNVRs) are rarely duplicated or lost in the mammalian lineage. Conversely, genes overlapped by benign variants have highly variable copy number across the tested species. Furthermore, we find that genes with conserved copy number across mammals are depleted among CNVs in non-human healthy mammals, mirroring the pattern observed in humans. These results demonstrate the role of dosage sensitivity in shaping the human genome and point to the usefulness of evolutionary metrics in refining the lists of candidate causative genes on pathogenic CNVs.

## Results

**Identification of pathogenic CNV peak regions**. We obtained human autosomal germline copy number gains (CNGs) and losses (CNLs) with clinical interpretations of 'benign' or 'pathogenic' from dbVar (Table 1). The operational definition of a CNV varies between studies, but in the data used here the minimum length of a CNV is 50 bp. Furthermore, we excluded CNVs that were >10% of the length of the respective chromosome, as these dramatically increase the number of genes included and potentially confound the analysis. Although benign CNVs outnumber pathogenic CNVs by about 2:1, the proportion of genome covered by any pathogenic CNV (74.4%) is much larger than that covered by benign CNVs (8.3%), due to the substantially longer average length of pathogenic CNVs.

CNVs are described by their start and end points, and whether they are gain or loss events (CNG and CNL, respectively). A given region of genome may be overlapped by multiple CNVs with different start and end points, of different types (gain or loss) and of different clinical interpretations. Sets of partially overlapping CNVs are grouped together into CNVRs. By contrast, other regions have no observed CNVs at all or only have rare CNVs.

The number of genes included in pathogenic CNVs seems implausibly large for them all to be causative of disease (Supplementary Figs 1 and 2). Rather, it is probable that only a subset of the genes in the pathogenic CNVRs are responsible for the associated phenotypes. We observe that 87.1% (223.8 Mb/256.9 Mb) of benign CNVR is overlapped by pathogenic CNVs. Thus, we wanted to refine the CNVs to home in on probable causative genes.

Even in well-characterized pathogenic CNVRs such as 22q11, the start and end points of the CNVR are variable between patients. However, a 'critical' 1.5 Mb region has been identified, which is common to most cases, and it is usually inferred that the primary causative genes are present within this region[38]. Similarly, Down's syndrome is caused by trisomy of chromosome 21, but a short segment of the chromosome has been linked to most symptoms and is considered the Down's syndrome critical region[39]. Mirroring this approach, we identified recurring subregions of pathogenic CNVRs as we consider them more likely to contain the causative genes.

We refined pathogenic CNVRs into 'peak regions' defined as local maximums of CNV coverage (Fig. 1). This approach has the advantage of promoting recurrent CNV subregions for special attention while also avoiding discriminating against rare CNVs in the data set; in cases where there is only one CNV in a region, the entire CNV is the local 'peak' (with a coverage of 1). We preferred this method to selecting an arbitrary genome-wide coverage

**Table 1 | Summary of human CNVs used in CNV analysis.**

| | # | Average length (kbp) | Combined length (Mbp) | Genome coverage | PC genes | PC developmental genes | Contain 1+ developmental genes |
|---|---|---|---|---|---|---|---|
| *Benign* | | | | | | | |
| CNVs | | | | | | | |
| Full | 8,005 | 388.5 | 3,110.1 | 8.9% | 1,802 | 318 | 28.0% (2,242) |
| Regions | 769 | 334.0 | 256.9 | | | | 27.1% (208) |
| Peaks | 993 | 128.3 | 127.4 | 4.4% | 1,108 | 223 | 18.8% (187) |
| CNGs | | | | | | | |
| Full | 4,306 | 400.1 | 1,723.0 | 6.2% | 1,369 | 216 | 26.5% (1,139) |
| Regions | 494 | 362.0 | 178.9 | | | | 28.7% (142) |
| Peaks | 619 | 82.9 | 94.9 | 3.3% | 875 | 159 | 20.4% (126) |
| CNLs | | | | | | | |
| Full | 3,699 | 375.0 | 1,387.1 | 4.2% | 822 | 148 | 29.8% (1,103) |
| Regions | 445 | 271.6 | 120.9 | | | | 21.1% (94) |
| Peaks | 506 | 143.9 | 72.8 | 2.5% | 555 | 112 | 17.2% (87) |
| *Pathogenic* | | | | | | | |
| CNVs | | | | | | | |
| Full | 4,366 | 3,503.0 | 15,294.2 | 80.3% | 16,343 | 3,742 | 95.4% (4,166) |
| Regions | 167 | 13,856.3 | 2,314.0 | | | | 92.8% (155) |
| Peaks | 923 | 545.0 | 503.1 | 17.5% | 4,234 | 1,117 | 58.2% (537) |
| CNGs | | | | | | | |
| Full | 1,097 | 3,985.1 | 4,371.6 | 48.4% | 11,217 | 2,512 | 97.2% (1,066) |
| Regions | 178 | 7,840.5 | 1,395.6 | | | | 92.1% (164) |
| Peaks | 300 | 1,861.6 | 558.5 | 19.4% | 4,365 | 1,025 | 76.7% (230) |
| CNLs | | | | | | | |
| Full | 3,269 | 3,341.3 | 10,922.6 | 67.7% | 13,128 | 3,058 | 94.8% (3,100) |
| Regions | 212 | 9,196.1 | 1,949.6 | | | | 92.0% (195) |
| Peaks | 699 | 669.7 | 468.1 | 16.3% | 3,653 | 999 | 63.7% (445) |

CNG, copy number gain; CNL, copy number loss; CNV, copy number variant; PC, protein coding.

threshold, as such an approach would exclude rare CNVs and low coverage regions, and would fail to refine high coverage regions. This is important, as some rare CNVs have been implicated in disease[14,40] and there are likely to be more, as yet uncharacterized, rare CNVs that are causative of disease.

Using this peak region approach, 167 CNVRs (composed of 4,366 individual CNVs, and grouping duplication and deletion CNVs), which cover over 74% of the genome and encompass 16,343 protein-coding genes were broken into 923 peak regions (a CNVR can have multiple local peaks) covering 16.2% of the genome and 4,234 genes (Table 1). Some of these peak coverage pathogenic regions overlap benign CNVs, intersecting with 16.7% (42.9 Mb/256.9 Mb) of benign CNVR.

A similar analysis can also be applied to benign CNVs (shown in Table 1 for comparison), but it makes little sense to analyse the benign CNVs in this way as we presume that the entire region is benign.

**Pathogenic CNVs are enriched for developmental genes.** CNVs have been associated with diverse conditions such as heart disease, cancers, immunodeficiency, hearing loss and obesity[8,17,18,41–47]; however, they are most often associated with developmental conditions; over 14% of developmental delay and intellectual disability cases are caused by CNVs[17]. This makes intuitive sense as development is considered to be a finely balanced, dosage-sensitive process[26,29]. Nonetheless, one must be careful to consider the possibility of an ascertainment bias: it is not possible to know whether a given individual will get heart disease later in life so they will be noted as healthy, whereas developmental conditions are early onset by definition and so should always be observed when present. Thus, it is not currently clear whether the apparent enrichment for developmental

conditions reflects a detection bias or a greater inherent vulnerability in developmental processes.

We found that 95.4% of full pathogenic CNVs in the current data set contain at least one developmental gene, compared with only 28.0% of benign CNVs. However, as pathogenic CNVs are typically longer and cover such a large proportion of the genome, it is expected that they will contain more genes and in turn are more likely to contain a gene involved in any given Gene Ontology (GO) category. Thus, it is necessary to correct for differences in CNV length. We did this by calculating the proportion of genes on each CNV that are developmental genes. When we considered individual pathogenic CNVs that were not overlapped by benign CNVs (that is, exclusively pathogenic regions), a mean of 37.3% of the genes were developmental genes compared with 24.2% of benign CNV genes (medians 28.4% and 0%, respectively), a highly significant difference ($P < 1.0 \times 10^{-16}$, Mann–Whitney $U$-test). As an alternative correction for length difference we randomized the location of the pathogenic CNVs and counted the number containing at least one developmental gene. We repeated this simulation 1,000 times. Over these simulations the mean percentage of CNVs that overlapped at least one developmental gene was 74.8% and the highest percentage found in any simulation was 76.9%, significantly less than observed in the real data ($P < 1 \times 10^{-16}$; Z-score: 37.2; Supplementary Fig. 3).

Comparing pathogenic peak regions to benign CNVRs (full merged CNVs), pathogenic regions that have no benign overlap were significantly enriched for containing at least one developmental gene (58.5% of 684 pathogenic regions versus 27.1% of 769 benign regions, $P < 1.0 \times 10^{-16}$, $\chi^2$-test). Although the lengths of full benign CNVRs (mean 334.0 kb; median 151.5 kb) and pathogenic peak region CNVRs (mean 545.6 kb; median 184.0 kb) are more similar than the full regions compared with

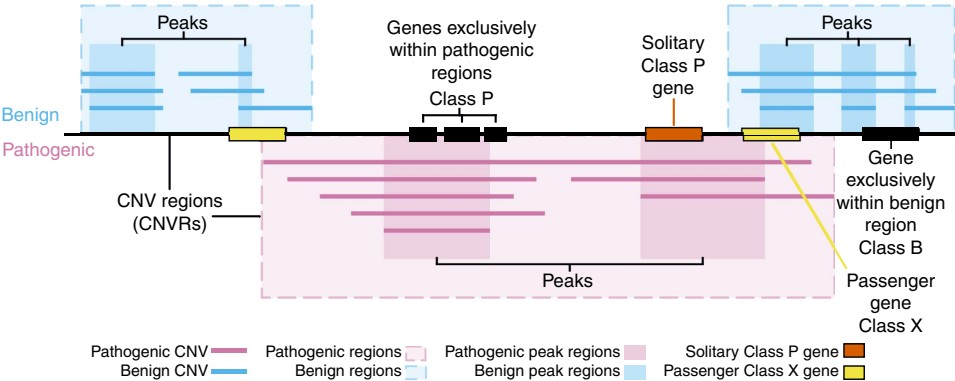

**Figure 1 | Illustration of CNVRs and intersection with genes.** Illustrative CNVs are shown with benign CNVs above the genomic region and pathogenic CNVs below (blue and pink lines respectively). Shaded boxes bound CNVRs with local peak coverage regions indicated by darker shading. Genes overlapped by both benign and pathogenic CNVs are termed Class X 'passenger' genes here (yellow). Where only a single non-passenger pathogenic gene is within a region, it is termed a 'solitary Class P' gene (orange).

each other (full pathogenic CNVRs: mean 13.9 Mb; median 8.3 Mb), the pathogenic regions still contain more genes on average. Thus, we corrected for gene number and found a mean of 33.7% developmental genes when we considered pathogenic peak CNVRs that were not overlapped by benign CNVRs compared with 24.5% of benign CNV genes (medians 20.0% and 0%, respectively), a highly significant difference ($P < 1.0 \times 10^{-16}$, Mann–Whitney $U$-test).

Clustering of developmental genes may contribute to their pathogenicity[48]. There is a significant enrichment of the proportion of developmental genes in pathogenic peak regions compared with pathogenic regions outside of peaks (pathogenic peak regions: 24.5% of 3,452 genes exclusive to these regions; remaining pathogenic regions: 18.1% of 15,978 genes; $P < 1.0 \times 10^{-16}$, $\chi^2$-test). We confirmed that this is not due to clustering of developmental genes in general in the genome, because on randomizing CNV location as above, the proportion of developmental genes covered by CNVs was consistently lower than observed in simulation (Supplementary Fig. 4). The fact that the peak regions are enriched for developmental genes with respect to the remainder of the the pathogenic CNVR is strong evidence that developmental genes are consistently implicated in CNV-related disease phenotypes across different CNVs in the genome.

**Class P genes display features of dosage-sensitive genes.** A particular region of genome can be overlapped by multiple CNVs and these may differ in both their type (gain or loss) or their clinical interpretation (benign or pathogenic). We consider genes that are in CNVs with opposite clinical interpretations unlikely to be causative, in particular when the CNV is of the same type. That is, a gene found in a benign CNV gain region and a pathogenic CNV gain region is unlikely to be driving the pathogenic phenotype (Fig. 1).

We refined the gene lists by considering all CNVs simultaneously. Figure 2a shows counts of genes according to CNV type and clinical interpretation. For benign CNVs we considered the full CNV, whereas for pathogenic CNVs we considered only peak regions as above. We identify 6,367 genes found only in pathogenic CNVs (shaded blue in Fig. 2b) and we label these 'Class P genes'. By contrast, 524 genes were inconsistent, reported in both benign and pathogenic CNVs, and are deemed to be benign 'passengers' (shaded grey in Fig. 2b; referred to as 'Class X'). We refer to the 1,075 genes that were consistently found in benign CNVs (shaded red in Fig. 2b) as Class B genes.

We tested these groups for enrichment of developmental genes, haploinsufficiency[49], protein complex members (Uniprot complex subunits[50]), ohnologues and high gene expression[51] (Table 2), all being features of genes previously associated with dosage sensitivity. We found Class B to be depleted for involvement in development and we found the opposite for Class P genes (14.3% versus 25.2%, $P = 2.5 \times 10^{-11}$, $\chi^2$-test). Using the probability of loss-of-function mutation intolerance as a proxy for probability of haploinsufficiency[49], we found Class B genes to be less likely to be haploinsufficient compared with Class P genes (median probability of loss-of-function intolerance: 0.014 versus 0.028, $P = 0.005$, Mann–Whitney $U$-test). In addition, considering only the subset of genes with high haploinsufficiency scores (3,230 genes with probability of loss-of-function intolerance $> 90\%$), we find Class P genes enriched (19.5%) relative to Class B and Class X genes (12.0% and 13.9%, respectively, $P = 4.3 \times 10^{-6}$, $\chi^2$-test).

Protein complex members are expected to have constrained relative dosages[52,53]. Although only 23.4% of Class B genes have products functioning as subunits in protein complexes, 33.9% of Class P genes are in complexes, a significant difference ($P = 7.6 \times 10^{-9}$, $\chi^2$-test). In addition, ohnologues (paralogues generated by WGD) are over-represented in Class P genes (37.6% versus 26.9%, $P = 3.9 \times 10^{-10}$, $\chi^2$-test), consistent with previous observations that ohnologues are frequently associated with disease (OMIM classification)[30]. There is evidence that highly expressed genes are not only strongly constrained with respect to sequence evolution[54–57] but also have greater dosage constraint[58]. Consistent with this we found that Class P genes have higher expression than Class B genes (medians: 19.6 reads per kilobase of transcript per million mapped reads (RPKM) and 9.6 RPKM, respectively, $P < 1.0 \times 10^{-16}$, Mann–Whitney $U$-test). Furthermore, Class P genes are more highly expressed than Class X genes (medians: 19.6 RPKM versus 12.6, $P = 4.7 \times 10^{-13}$, Mann–Whitney $U$-test). The trends observed here are consistent with the notion that genes in pathogenic CNVs are dosage sensitive.

**Solitary Class P genes are enriched for neurodevelopment.** When we consider the genomic distribution of the Class P genes we observe that 7 of 390 CNVRs (178 pathogenic CNG regions and 212 pathogenic CNL regions) do not contain any genes exclusive to pathogenic CNVRs. In these cases, pathogenicity may be due to genes of reduced penetrance, position effects of the CNV or a different type of dosage sensitivity (for example, if the

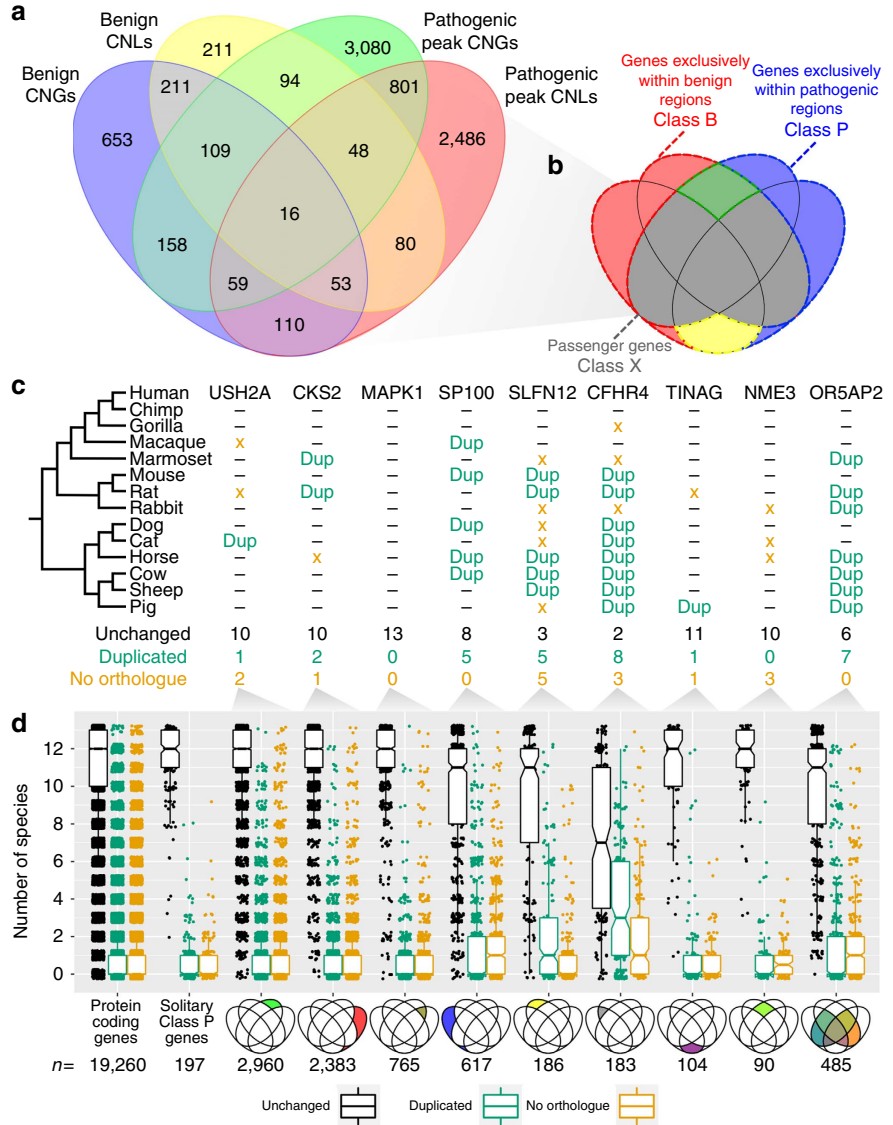

**Figure 2 | Patterns of gene duplication and loss across mammals for orthologues of human genes in CNVs.** (**a**) Venn diagram showing the number of protein-coding genes overlapped by different combinations of CNV types (blue, benign CNGs; yellow, benign CNLs; green, pathogenic gain peak coverage regions; red, pathogenic loss peak regions). (**b**) Genes that are covered exclusively by benign CNVs are labelled as 'Class B' (shaded red), those exclusive to pathogenic CNVs as 'Class P' (shaded blue) and those falling in CNVs with both clinical interpretations for gain or loss are considered as likely to be passenger genes and labelled 'Class x' (shaded grey). It is noteworthy that the classification refers to the CNVs that the genes fall within rather than the genes themselves. (**c**) Phylogenetic tree of 13 mammalian species used for gene conservation analysis and examples of human genes from each CNV overlap pattern type (Venn diagram segment) showing the orthologue distribution in the mammals. A dash indicates no change. (**d**) Box plot of the number of mammalian species where copy number is unchanged (black), duplication has occurred (green) and no orthologues (orange) for different categories of CNV overlap, as indicated below the boxplots. Upper and lower hinges of boxes correspond to the first and third quartiles. The median is shown within each box. Whiskers extend to values 1.5 × interquartile range. These data were calculated per gene as illustrated in **c**. The sample size is shown below each boxplot.

gene is haploinsufficient, or conversely if the gene is aggregation prone at higher concentration, then these genes could be in both pathogenic loss and benign gain CNVs or vice versa, respectively, and would not be designated as exclusively pathogenic by us).

The remainder of CNVRs contain at least one protein-coding gene that is never observed in a benign CNV. We found that 21/390 pathogenic CNVRs contain exactly one such Class P gene and 300/999 pathogenic peak regions contain exactly one (199 unique genes out of 321 solitary genes intersecting with regions; Supplementary Data 1 and Supplementary Figs 5 and 6). These latter cases have suggested pathogenicity by exclusion: they

are only found in pathogenic CNVs and no other gene in the peak region is exclusively pathogenic. The observation that peak regions are enriched for solitary pathogenic genes suggests that analysing peak regions is a useful way to refine the analysis of CNVRs. It is noteworthy that we consider sets of overlapping duplication CNVs separately from deletion CNVs when building these CNVRs and their peak regions. This allows for the possibility of the mechanistic basis of pathogenicity being different between duplication and deletion CNVs, although frequently it is the same gene that is the sole Class P gene: 122 out of 199 solitary Class P genes are the solitary Class P gene for a duplication and a deletion CNV peak region.

**Table 2 | Genes included in different types of CNV have different genetic and functional characteristics.**

| | Class B genes (1,075)* | Class P genes (6,367)† | Class X genes (523)‡ | BL/PG genes (94) | BG/PL genes (110) | P-value (χ²-test)§ | P-value (Mann–Whitney U-test)‖ |
|---|---|---|---|---|---|---|---|
| Developmental genes | **14.3% (154)** | **25.2% (1,606)** | 22.4% (117) | 20.2% (19) | 25.5% (28) | $2.7 \times 10^{-11}$ | |
| Protein complex members | **23.4% (251)** | **33.9% (2,156)** | **28.7% (150)** | 33.0% (31) | 35.5% (39) | $7.5 \times 10^{-9}$ | |
| Ohnologues | **26.9% (289)** | **37.6% (2,395)** | **30.4% (159)** | 29.8% (28) | 41.8% (46) | $4.1 \times 10^{-10}$ | |
| Haploinsufficient genes¶ | **12.0% (104)** | **19.5% (1,138)** | **13.9% (63)** | 13.4% (11) | 14.9% (15) | $4.3 \times 10^{-6}$ | |
| Haploinsufficiency score (median)# | 0.014 | 0.028 | 0.009 | 0.002 | 0.001 | | |
| | • | • | | | | | 0.005 |
| Maximal expression in RPKMs (median) | 9.6 | 19.6 | 12.6 | 20.4 | 14.1 | | |
| | • | • | | | | | $<1.0 \times 10^{-16}$ |
| | • | | | | • | | 0.005 |
| | | • | • | | | | $4.5 \times 10^{-13}$ |

BG/PL, genes exclusively overlapped by benign gain CNVRs and pathogenic loss peak CNVRs; BL/PG, genes exclusively overlapped by benign loss CNVRs and pathogenic gain peak CNVRs; CNV, copy number variant; CNVR, CNV regions; RPKM, reads per kilobase of transcript per million mapped reads.
*Genes exclusively observed in benign CNVRs.
†Gene exclusively observed in pathogenic CNVRs.
‡Genes observed in contradictory CNV types and clinical interpretations.
§All P-values are Bonferroni corrected. Values in bold have adjusted residuals > ± 2 in the χ²-test.
‖Pairwise comparisons are indicated with dots. All P-values are Bonferroni corrected.
¶Genes with probability of loss-of-function mutation intolerance > 90% inferred in ref. 49.
#Probability of loss-of-function mutation intolerance inferred in ref. 49.

Thus, of the 4,234 genes overlapped by pathogenic peak region CNVs, 199 are the solitary candidate pathogenic gene in the region. These are promising candidates for causing the pathogenicity of the CNV. With all pathogenic peak CNVR genes as a background list, we find that solitary candidate pathogenic genes are enriched for 'anatomical structure development' (GO:0048856; $P = 8.4 \times 10^{-12}$), especially 'embryonic morphogenesis' (GO:0048598; $P = 1.0 \times 10^{-10}$) and 'neurogenesis' (GO:0022008; $P = 3.1 \times 10^{-8}$), 'regulation of multicellular organismal process' (GO:0051239; $P = 3.7 \times 10^{-9}$), 'adult behaviour' (GO:0030534; $P = 4.7 \times 10^{-9}$) and 'signalling' (GO:0023052; $P = 3.4 \times 10^{-8}$) (Supplementary Data 2). These 199 genes are also significantly enriched for localization within axons and dendrites (cellular component term 'neuron projection', GO:0043005; $P = 8.9 \times 10^{-6}$) and are overrepresented for genes associated with an 'abnormality of the nervous system' (HP:0000707; $P = 4.7 \times 10^{-30}$) in the Human Phenotype Ontology[59].

Although representing only a small portion of all genes overlapped by pathogenic CNVs, solitary non-passengers are overrepresented for clinically relevant functional categories.

**Class P genes have high evolutionary copy number constraint**. Under the hypothesis of CNV pathogenicity being caused by the dosage sensitivity of enclosed genes, we expect to see characteristic patterns of evolution of genes within pathogenic CNVs, namely a dearth of gene duplication and loss events. We investigated gene duplication and loss within the mammalian tree by counting the number of genomes in which there are copy number changes. For a given human gene that was inferred to have been present in the mammalian common ancestor (that is, excluding newer genes and genes where the orthologue is not identifiable). We looked across 13 genomes and noted whether there was a gene duplication, absent orthologue or no change in that genome (for examples, see Fig. 2c).

We performed this for all human genes present in the mammalian ancestor and grouped the results according to the presence of the human gene in benign or pathogenic, gain or loss CNVs as before. The box plots of these distributions are shown in Fig. 2d. Panels 2, 3, 4 and 5 show that the conservation of copy number for genes in pathogenic regions is high across the

genomes surveyed (the copy number is mostly unchanged, black points; median 12, with upper and lower quartiles at 13 and 11 species). By contrast, for the genes in benign, presumably less dosage-sensitive regions, the copy number is more variable, with a greater proportion of genes having copy number changes in more genomes (lower quartile ranging between eight and four species). These genes show more duplications and missing orthologues across the mammalian tree. Variance significantly increases from pathogenic groups to benign groups ($P < 1 \times 10^{-16}$, Fligner–Killeen test) indicative of lower copy number constraints in the latter. We also compared the counts of genes with conserved copy number in all tested species in each CNV classification group (Fig. 2d panels 3–11) and found a highly significant difference ($P < 1 \times 10^{-16}$, χ²-test).

This evolutionary analysis provides an independent measure of gene dosage sensitivity and is not dependent on CNV classification, yet the patterns match the expectations based on CNV clinical interpretations, namely genes with evolutionary patterns suggestive of dosage sensitivity are more associated with pathogenic CNVs.

**Evolutionary copy number conservation in pathogenic CNVRs**. If we imagine a simplified scenario where there is a single dosage-sensitive gene in a region, the observed peak CNVR may nonetheless repeatedly contain multiple genes. This will be particularly true in the case of CNV hotspots[10], which may be located at several genes' distance, repeatedly generating multi-gene CNVs. In this scenario, neither the dosage-sensitive gene nor the closely linked non-dosage-sensitive genes will be observed in benign CNVs. Similarly, as evolutionary gene duplication events have the same mechanistic origins as CNVs, linked non-dosage-sensitive genes may have patterns of duplication and loss that somewhat track the pattern of the dosage-sensitive gene. However, if the linkage is broken by genome rearrangement events, this incidental constraint on the non-dosage-sensitive gene will be broken. Thus, genes with the most consistent patterns of gene copy number conservation are the most interesting.

We applied the evolutionary constraint metric to genes within 14 well-characterized pathogenic regions associated with neurodevelopmental disorders. Figure 3a shows the copy number

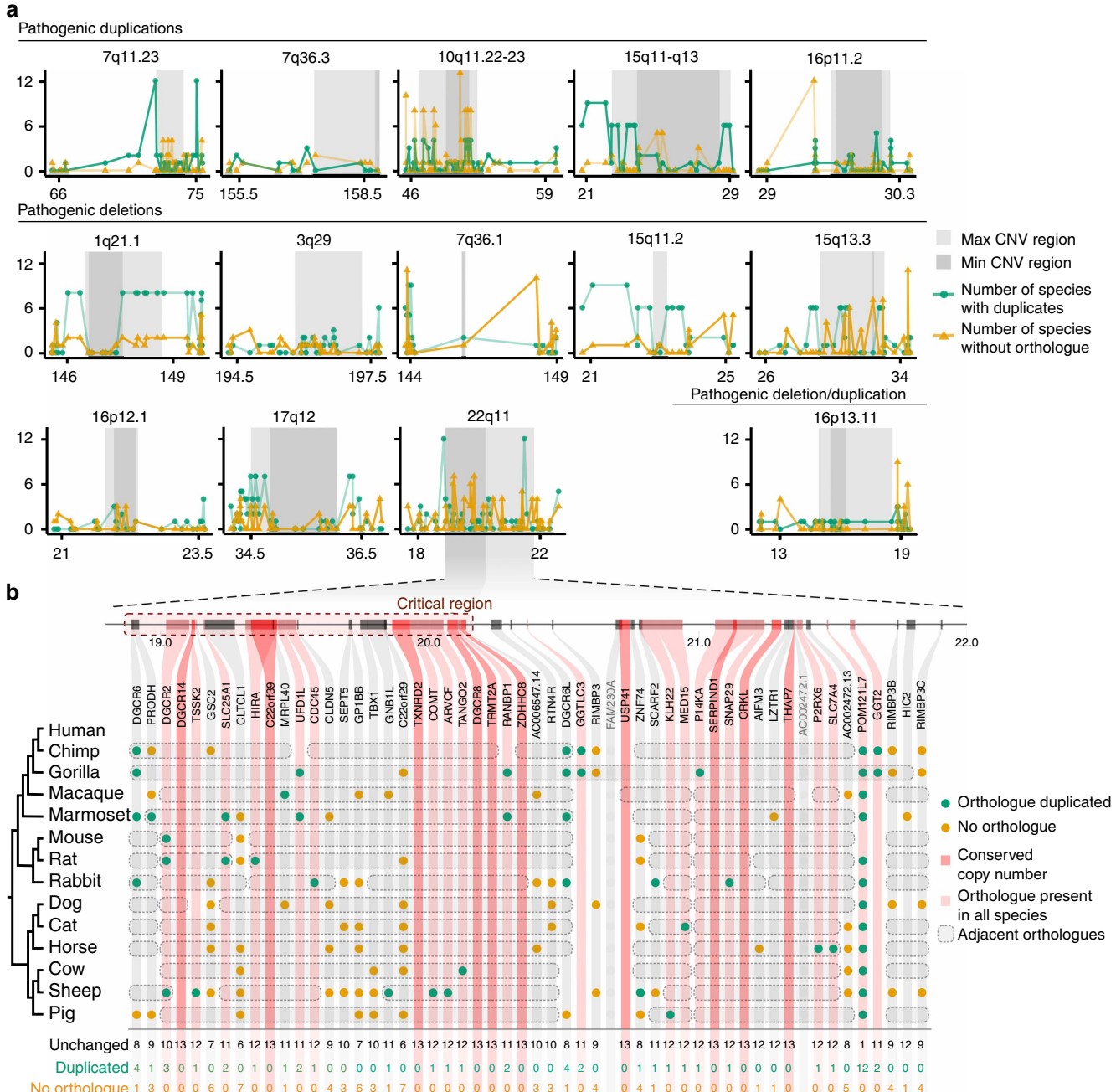

**Figure 3 | Mammalian copy number changes for genes within known pathogenic CNVRs.** (**a**) Copy number changes across mammalian species for genes within known pathogenic CNVRs associated with schizophrenia and other neurodevelopmental disorders obtained from ref. 36. The minimal (min) CNVR (shaded dark grey) is typically the smallest region associated with the disease phenotype, while the maximal (max) CNVR (shaded light grey) is typically observed. Ten flanking genes on each side are also plotted where possible. Each region is labelled above with the chromosomal band and position along chromosome in megabases is shown on the x axis. Genes are plotted by start position. Each point represents for one human gene the number of duplications (green) and losses (orange). Genes within regions are listed in Supplementary Data 6. (**b**) For each protein-coding gene within the 22q11 region, copy number changes across 13 mammalian species are shown. Green circles indicate where orthologues are duplicated, orange circles where orthologues are missing. Genes highlighted in light red are genes where at least one orthologue is present in all species and genes highlighted in dark red are genes with conserved one-to-one orthology across the mammalian species tested (completely conserved genes). Grey dashed outlines group orthologues that are neighbouring on their respective chromosome/scaffold in each species. Genes with greyed-out names were not included in copy number analysis and so no data are displayed for them.

variation across mammals of genes in each of these regions along with some flanking genes. The flanking regions are included as a proxy indicator of any potential local duplication or loss biases. In some cases, such as 1q21.1 deletion and the 17q12 deletion, the pattern of gene conservation across mammals fits expectations in

that the genes within the pathogenic CNVR are much more conserved than the genes in the flanking regions. Other regions have patterns of conservation that are close to expectations and others show no obvious gross pattern. There is no significant difference between the total pathogenic CNV and the flanking

region. However, when we compare the medians of number of 'unchanged' gene copy number inside the critical region (shaded dark grey, when present) with the medians in the remainder plus the flanking region, we find that the genes in the critical region are significantly more evolutionarily conserved (Mann–Whitney $U$-test, Bonferroni-corrected $P = 0.0028$).

One of these is the region associated with 22q11 deletion syndrome, which is shown in detail in Fig. 3b. Even though Fig. 3a shows several genes with multiple loss events, the detailed view shows that ten of the genes in this region have no duplication or loss events in any of the mammalian genomes tested (dark red shading). Of the 28 genes present in the critical region of this CNV (1.5 Mb deletion that presents the same symptoms as the 3 Mb deletion[38]), 16 have consistently detectable orthologues in all mammalian genomes and three are missing an orthologue in only one of the 13 genomes. We chose these 13 genomes for analysis based on the high quality of the available data, but even so, we cannot exclude that some of these differences are due to missing data or poor annotation. Nonetheless, the subset of 22q11 genes that are completely conserved across mammals are good candidates for disease causation. Interestingly, TBX1, a candidate disease gene in this syndrome[60] is not completely conserved, not being detected in cow, sheep and pig. This is consistent with a single loss event in this more distant mammalian lineage, which may indicate differing constraints in these mammals.

**Conserved genes reveal ancient and persistent constraints.** We identified 7,014 human genes that have conserved copy number across all 13 mammalian genomes (Supplementary Data 3). Even though this definition is independent of CNV status, this evolutionary information is suggestive of dosage constraint. Over 28% of these are involved in development, consistent with genes identified via pathogenic CNVs. Overall, we found that the evolutionarily conserved genes are strongly enriched for 'anatomical structure development' (GO:0048856; $P = 1.7 \times 10^{-31}$) as part of developmental processes (GO:0032502; $P = 2.2 \times 10^{-30}$), 'cell communication' (GO:0007154; $P = 3.4 \times 10^{-31}$), 'phosphorus metabolic process' (GO:0006793; $P = 4.3 \times 10^{-27}$) and 'macromolecule modification' (GO:0043412; $P = 5.3 \times 10^{-23}$) specifically 'protein modification process' (GO:0036211; $P = 2.5 \times 10^{-23}$).

In addition, conserved genes are also enriched for 'localization' (GO:0051179; $P = 2.8 \times 10^{-23}$), 'regulation of biological process' (GO:0050789; $P = 3.3 \times 10^{-21}$) and 'response to stimulus' (GO:0050896; $P = 3.5 \times 10^{-21}$), encompassing 'response to organic substance' (GO:0010033; $P = 6.7 \times 10^{-16}$), 'response to endogenous stimulus' (GO:0009719; $P = 7.0 \times 10^{-12}$) and 'response to oxygen-containing compound' (GO:1901700; $P = 1.8 \times 10^{-10}$) (Supplementary Data 4). We confirmed that the enrichment for developmental genes is not due to clustering of those genes on the genome, because we observe no effect of distance to the nearest developmental gene in the human genome and the conservation across mammalian genomes (Supplementary Fig. 7). Furthermore, we observed that genes conserved across the mammalian tree are enriched for OMIM disease genes (18.2%;$P < 1.0 \times 10^{-16}$, $\chi^2$-test) relative to genes with copy number changes (13.4%). We find a similar trend for candidate haploinsufficient genes (genes with probability of loss-of-function mutation intolerance > 90% inferred in ref. 49) with conserved genes enriched (22.8%) compared with genes with copy number changes (15.0%; $P < 1.0 \times 10^{-16}$, $\chi^2$-test). Clearly, conserved genes are functionally distinct and involved in biologically important processes.

We tested genes with conserved copy number for representation among genes overlapped by benign CNVs and genes overlapped by an independent human CNV map[9]. We found them to be underrepresented among benign CNV genes with 5.8% (393/6,809) of conserved genes overlapped by benign CNV compared with 10.9% (1,272/11,632) of genes not conserved in all 13 genomes tested ($P < 1.0 \times 10^{-16}$, $\chi^2$-test). Similarly, conserved genes are overlapped less in a control CNV map (35.6% of 7,014 conserved genes overlapped) compared with genes not conserved in the genomes tested (38.4% of 13,300 genes, $P = 0.0001$, $\chi^2$-test). This is consistent with our expectation that these genes are under copy number constraint and with previous work that has shown comparatively more duplications of genes in benign CNVRs[61] and of haplosufficient genes[62].

The pattern of conservation across the mammalian tree suggests an ancient and persistent dosage constraint, and as such we expect that CNVs encompassing orthologues of these genes would also be deleterious in other mammals. We tested mouse orthologues of genes with conserved copy number and we found them to be depleted among mouse CNVs compared with other protein-coding genes (23.6% of mouse orthologues of conserved genes are overlapped by mouse CNVs compared with 27.3% of other mouse genes, $P = 3.0 \times 10^{-9}$, $\chi^2$-test). This indicates that these genes are constrained compared with other genes within mouse. These results suggest that evolutionary trends are informative in the identification of dosage-sensitive genes.

## Discussion

Although the phenotypes resulting from CNVs at different genomic locations can differ quite widely, there are certain commonalities that allow a deeper insight into the genetic and biological mechanisms of CNV pathogenicity. The fact that we observe trends in function and evolutionary patterns for genes within pathogenic CNVs supports the hypothesis that gene dosage sensitivity is a predominant causative factor. In particular, CNV subregions that recur frequently in pathogenic cases, or CNVRs that are rare but associated with pathogenicity, are biased with respect to the genes they contain both in terms of function and evolution.

In particular, the observation that genes with constrained evolutionary patterns of gene duplication and loss are usually found within pathogenic CNVs strongly supports the model whereby dosage sensitivity of individual genes enclosed by a CNV is responsible for pathogenicity. This pattern is not predicted by any other model (although does not exclude the co-existence of other mechanisms of CNV pathogenicity). Furthermore, the identification of genes with such evolutionary patterns supplies a shortened list of candidate genes for further inspection. Peak regions of pathogenic CNVs that contain only one gene that is exclusively found in pathogenic CNVs are of particular interest. Based on an admittedly simplistic logic, these 199 genes are candidate causative disease genes. Consistent with this, these genes are rarely found to be duplicated or lost in other mammals (Fig. 2d, panel 2).

Importantly, this analysis of gene duplication and loss is restricted to genes where we can infer the presence in the common ancestor to all 13 mammalian genomes examined. Thus, we avoid any problems associated with the increased difficulty in detecting quickly evolving genes[63,64]. Genes that we cannot infer to be present in the common ancestor are either new genes or older genes that are difficult to detect because of gene loss or extensive sequence evolution and it is not possible to distinguish these without more detailed inspection of the loci. However, we found that genes that were not inferred in the ancestral mammal are enriched in benign CNVRs compared with the rest of the genome (7.6% (137/1,802) versus 4.8% (852/17,628) respectively, $P = 4.7 \times 10^{-7}$, $\chi^2$-test), suggesting lower evolutionary

constraint, consistent with having less phenotypic effect on disruption.

Haploinsufficient genes are genes where there is a minimum amount of gene product required to attain a wild-type phenotype. Logically, these are distinct from dosage-balanced genes where any significant disruption in amount of product, be it increased or decreased, will induce a phenotype; however, in practice the two may overlap (if, for example, one tests only for a phenotype in heterozygote knockouts). Interestingly, in their analysis of haploinsufficiency, Huang et al.[62] observed fewer paralogues of haploinsufficient genes, even though this is not predicted by haploinsufficiency, but which would be expected of general dosage sensitivity or dosage balance. We would expect that the genes showing the pattern indicated by the yellow segment in Fig. 2b, which is benign gain but pathogenic loss, should naturally be haploinsufficient genes. Conversely, genes present in pathogenic gain CNVs but benign loss CNVs (green segment in Fig. 2b) may be aggregation prone at high concentration. Whereas we lack well-curated data on aggregation-prone genes to test the latter relationship, we can use the recently available haploinsufficiency data to test the former. We observed the expected enrichment for haploinsufficiency among genes found within benign gain, pathogenic loss regions. These might be considered 'simple' haploinsufficient genes. However, the enrichment among class P genes described above suggests that many haploinsufficient genes are also dosage sensitive in other ways.

There is great interest in the relationship between development and dosage sensitivity, and CNVs in general. As we noted, there is a potential bias in the annotation of disease CNVs due to this interest and due to the fact that developmental disorders are expected to be more reliably identified at the time of sample collection. Therefore, the enrichment for developmental genes in pathogenic CNVs must normally be interpreted in that light. However, the evolutionary measures based on conservation of copy number across mammalian species are independent of disease annotation and have no such reporter or study bias. Our finding that these evolutionarily constrained genes are indeed enriched for developmental genes confirms the view of development as an inherently dosage sensitive process.

This is the first comparison of the genome evolutionary trends of genes in benign and pathogenic CNVs. We have revealed distinct functional and evolutionary trends for the two classes of CNVs. This points to the usefulness of evolutionary metrics in the interpretation of CNVs.

## Methods

No statistical methods were used to predetermine sample size. There was no randomization of experiments. Investigators were not blinded to group allocation during experiments and outcome assessments.

**CNV data with clinical interpretation.** Human autosomal germline CNVs with clinical interpretations of 'benign' and 'pathogenic' were obtained from dbVar release dated 31 October 2013 for genome assembly GRCh37 (ref. 65). CNVs longer than a tenth of a chromosome were discarded and the included CNVs coordinates are listed in Supplementary Data 5 and summarized in Table 1. dbVar studies included in this analysis are listed in Supplementary Table 1.

**CNV coverage.** CNV coverage (number of CNVs overlapping a given region of genome) was calculated genome-wide using Bedtools[66]. Within each CNVR, peak regions were calculated as any local maximum in CNV coverage, defined as any subregion with higher coverage than its flanking regions. Multiple peak regions were permitted within a CNVR. Where a given region has only one CNV, the entire CNV is counted as the peak region. Protein-coding gene annotations and GO terms were obtained from Ensembl GRCh37 (ref. 67). A gene was considered to be intersecting with a CNV if any of the gene sequence was overlapped by one or more bases on either strand.

**Gene category enrichment.** Median RPKM values by tissue were obtained from GTEx V6 (ref. 51). The tissue with the maximum RPKM value was used for each gene. Probability of loss-of-function mutation intolerance values for genes were obtained from ExAC Release 0.3 (ref. 49) as a proxy for haploinsufficiency scores. Protein complex member genes were sourced from the Uniprot KB/Swiss-Prot database[50] by filtering for keywords in the 'Subunit structure' annotation field.

**Mammalian copy number analysis.** Gene duplications and losses in 13 mammalian genomes (*Bos taurus*, *Callithrix jacchus*, *Canis lupus familiaris*, *Equus caballus*, *Felis catus*, *Gorilla gorilla*, *Macaca mulatta*, *Mus musculus*, *Oryctolagus cuniculus*, *Ovis aries*, *Pan troglodytes*, *Rattus norvegicus* and *Sus scrofa*) were calculated for all human genes inferred to be present in the mammalian common ancestor. Gene duplications and losses were inferred from Ensembl Compara annotations[67]. For each of the 13 species, a given human gene was considered to be duplicated in that genome if the annotation was one-to-many. We also counted the number of instances were the Ensembl Compara annotation reports no orthologue in that genome as presumed gene loss events. Genes with a one-to-one orthologous relationship were counted as unchanged in that genome. Where genes have apparently experienced a duplication event on a branch leading to humans since the mammalian divergence this could potentially confound the counts, because all genomes not sharing the duplication would be counted as being changed with respect to human. Thus, these recent human paralogues were grouped and their ancestral copy number of 1 was compared with the copy number of each other species.

**22q11 conserved synteny analysis.** For Fig. 3b, grey dashed outlines show orthologue groups that are neighbouring on their respective chromosome/scaffold in a given species. When no orthologue is present for a gene but orthologues exist for flanking genes and are neighbouring, one bounding outline groups all genes. This permits grouping in cases where additional genes are annotated within orthologous sequence in one of the genomes or where pseudogenization has occurred. Neighbouring groups are broken when chromosome/scaffold changes, region inversion occurs breaking gene collinearity or where position shifts substantially on the same chromosome/scaffold to a non-neighbouring region.

**CNV data for comparative genomics.** For comparative analysis of CNVs between species, human CNV data without clinical interpretation (presumed healthy) were obtained from the inclusive map provided in ref. 9. Mouse CNV data for wild-caught mice from four populations were obtained from ref. 5. There is no pathogenicity information explicitly listed, but they are presumed to represent healthy control variation. CNVs are identified on every mouse chromosome. Mouse orthologues of human genes with one-to-one relationships in all 13 mammalian species tested were intersected with mouse CNVs. Genes overlapped by one or more bases were considered to be affected by CNVs.

**GO enrichment analysis.** Developmental genes were defined as those with GO term 'developmental process' (GO:0032502).

GO term enrichment of solitary non-passenger pathogenic genes was examined using g:Profiler[68] with genes within full pathogenic CNVRs as a custom background gene list. The significance threshold was adjusted by Bonferroni correction. Genes that show one-to-one orthology in all 13 mammalian species were tested for GO enrichment compared with the full list of human protein-coding genes using g:Profiler.

**Code availability.** Custom scripts are available at https://github.com/alanrice/paper-dosage-sensitivity-copy-number-variation.

**Data availability.** The data reported in the paper are obtained from publicly available data repositories. Human autosomal germline CNVs were obtained from dbVar release dated 31 October 2013 for genome assembly GRCh37 (ref. 65). Protein-coding gene annotations and GO terms were obtained from Ensembl GRCh37 (ref. 67). Expression RPKM values were obtained from GTEx V6 (ref. 51). Probability of loss-of-function mutation intolerance values for genes were obtained from ExAC Release 0.3 (ref. 49) as a proxy for haploinsufficiency scores. Protein complex member genes were sourced from the Uniprot KB/Swiss-Prot database[50] accessed 14 September 2016. GO term enrichment was examined using g:Profiler[68]. rev. 1,270, for Ensembl 75. Mouse CNV data were obtained from ref. 5. Human control CNV data were obtained from the inclusive map provided in ref. 9. Gene duplications and losses were inferred from Ensembl Compara annotations[67]. Known pathogenic CNVRs associated with schizophrenia and other neurodevelopmental disorders were obtained from ref. 36. All additional data are available from the authors on reasonable request.

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

## Acknowledgements

We thank all members of the McLysaght research group for valuable discussions. This work is supported by funding from the European Research Council under the European Union's Seventh Framework Programme (FP7/2007–2013)/European Research Council grant agreement 309834.

## Author contributions

A.McL. and A.M.R. conceived the study. A.M.R. carried out experiments. A.M.R. and A.McL. analysed the data and wrote the manuscript.

## Additional information

**Competing financial interests:** The authors declare no competing financial interests.

**Publisher's note**: 

