## [Peer Review File · Nature Communications]

Reviewers' Comments:

Reviewer #1 (Remarks to the Author)

In this MS Rice and McLysaght have explored some characteristics of protein-coding genes within pathogenic CNVs (copy-number variations) in humans.

Criticisms/Comments (in no particular order of importance unless indicated):

1. The categorization of CNVs as pathogenic, and benign is well explained and illustrated (fig1). It is not clear however what is the extend of overlap between the two, and how the genes in the overlap were treated in each subsequent analysis. This reviewer suggests to present the results with and without these overlap regions.
2. pg5. please make clear that the 95.4% of pathogenic and only 28% of benign fractions of CNVs with at least one developmental gene are CORRECTED for the genomic size; same for the 58.2% and 27.1% of the last para of pg5.
3. pg6. This reviewer strongly suggests against the names pathogenic, passengers, and benign genes. The same genes may change categories for point mutations (LOF, non-synonymous substitutions etc), and for a variety of phenotypic traits. The authors may wish to use class1, 2, and 3 and clearly explain the different classes.
4. pg7. Please provide the definitions and metrics of the haploinsufficient genes, and genes on protein complexes. Ref44 for the haploinsufficient genes is dated and newer metrics for haploinsufficiency have been proposed. For genes in protein complexes, please look at the recent proteomic literature for lists.
5. pg7. 390 CNVRs while on pg 5 they are 167 CNVRs. Please make clear when the peaks are used and when the total regions are used. For pathogenicity I suggest to use only the peak regions
6. pg7. Regarding gene expression levels, I suggest to use the GTEX public data. Genes have different levels of expression in different organs and these could range for more than a log10 per gene in the different organs. I suggest to take the tissue with the highest RPKMs.
7. Is there any difference of old versus young genes ?
8. pg10. More info on the mouse CNVs is needed. Which regions? From which ref? Is there any pathogenicity attached to the mouse CNVs?
9. Fig3a. Is there any pvalue attached to each of the pathogenic duplications/deletions ? What is the null expectation for each one?

Reviewer #2 (Remarks to the Author)

This study examines the nature of pathogenic and benign copy number variants in humans. The findings reveal that those that are pathogenic contain genes that are dosage sensitive, involved with various developmental processes and evolutionarily constrained while benign cases are not similarly enriched for these classes of genes. The paper is well written and clear; the results seems reasonable and the arguments sound.

Some minor typos/clarifications:

In the section entitled "Pathogenic CNV are enriched for developmental genes", in the second line a semicolon should follow the citation brackets and a comma should follow "however".

In the second to last paragraph of the Discussion, it is stated that haploinsufficient genes are logically distinct from dosage balanced genes but notes that they are often conflated. It seems that haploinsufficient and dosage balanced genes could in fact overlap (rather than being "conflated") because they might both produce a phenotype from a heterozygous knockout. Semantics, yes, but maybe this could be rephrased?

Reviewer #3 (Remarks to the Author)

Rice and McLysaght present a study of genes found within pathogenic CNVs and their evolutionary constraints based on those submitted in dbVar. This is in general an interesting topic, as, despite many CNVs are reported as being pathogenic, the underlying gene(s) causing the phenotype is often still unknown.

Identifying key features of genes within these CNVs could contribute the identification of specific disease genes. After reading this manuscript I have a few suggestions.

Major:

1. As the authors mention the vast majority of pathogenic CNVs (submitted to dbVar) have till now been identified in patients with a neurodevelopmental disorder; largely due to current diagnostic strategies. This is in contrast to the knowledge that pathogenic CNVs cause a much broader range of phenotypes (for example, but not limited to, Glessner 2014, Orange 2011, Shearer 2014). This bias in the input dataset, likely to impact the results presented here, in particular the finding that the pathogenic CNVs are enriched in developmental genes, as the majority of the patients are affected by a developmental disorder. Either the conclusions from this result need to be reassessed ie. that pathogenic CNVs in developmental disorders are enriched for developmental genes, or the impact of the bias on the list of CNVs determined.
2. The authors conclude that genes found with in pathogenic CNVs are evolutionary conserved in copy number, these findings should be discussed within context of previous results on evolutionary forces within CNVs, for example Nguyen et al Genome Research 2008 and Huang et al Plos Genetics 2010.
3. p6 "Clustering of developmental genes may contribute to their pathogenicity". This concept will influence the results presented here. More analysis of this concept independent to the pathogenic CNVs is required.

Minor:

1. p12 Methods, more insights should be given on the datasets included in the study, it is unclear the range of data which has been included, patient numbers, study cohorts...
2. Table 1, requires more explanation in terms of footnotes, for example CNL, CNG
3. More details are required of how "developmental genes" are defined, it is currently unclear how this set has been defined.
4. Table 2, abbreviations should be more clearly explained. For example BL/PG
5. p5 final paragraph "Although the lengths of full benign CNV...are more similar"; this should be quantified.
6. Figure 2d, the sample size (n) should be clearly stated.

Reviewers' comments:

Reviewer #1 (Remarks to the Author):

In this MS Rice and McLysaght have explored some characteristics of protein-coding genes within pathogenic CNVs (copy-number variations) in humans.

Criticisms/Comments (in no particular order of importance unless indicated):

1. The categorization of CNVs as pathogenic, and benign is well explained and illustrated (fig1). It is not clear however what is the extend of overlap between the two, and how the genes in the overlap were treated in each subsequent analysis. This reviewer suggests to present the results with and without these overlap regions.

Response: For full benign CNVRs, 87.1% of their length is overlapped by pathogenic CNV. When only pathogenic peak regions are considered, 16.7% of the length of benign CNVRs is overlapped. These figures have now been added to the manuscript.

Additionally, throughout the text we have made many small edits to sentences

and paragraphs to clarify how overlaps were treated in each instance.

2. pg5. please make clear that the 95.4% of pathogenic and only 28% of benign fractions of CNVs with at least one developmental gene are CORRECTED for the genomic size; same for the 58.2% and 27.1% of the last para of pg5.

Response: In fact, the 95.4% refers to the uncorrected analysis, and the correction follows in the subsequent sentences. As the current text caused confusion, we tried to make this more clear. The current text reads:

“We found that 95.4% of full pathogenic CNVs in the current dataset contain at least one developmental gene, compared to only 28.0% of benign CNVs. However, as pathogenic CNVs are typically longer and cover such a large proportion of the genome it is expected that they will contain more genes and in turn are more likely to contain a gene involved in any given GO category. Thus it is necessary to correct for differences in CNV length. We did this by calculating...”

3. pg6. This reviewer strongly suggests against the names pathogenic, passengers, and benign genes. The same genes may change categories for point mutations (LOF, non-synonymous substitutions etc), and for a variety of phenotypic traits. The authors may wish to use class1, 2, and 3 and clearly explain the different classes.

Response: we have renamed these as Class P, Class B, and Class x for genes in pathogenic, benign and inconsistent CNVs respectively. This now appears in figure 1 as well as throughout the text where appropriate. We found that using class 1, 2, 3, etc. necessitated an explanation of each term upon each instance of use. We thought that P, B, and x retain some intuitive comprehensibility without misleading readers. In the description in the figure legend we make it clear that the P, B and x, refer to the clinical interpretation of the CNV that includes the gene, rather than an interpretation of the gene itself.

4. pg7. Please provide the definitions and metrics of the haploinsufficient genes, and genes on protein complexes. Ref44 for the haploinsufficient genes is dated and newer metrics for haploinsufficiency have been proposed. For genes in protein complexes, please look at the recent proteomic literature for lists.

Response: We obtained a recent set of candidate haploinsufficient genes based on human population sampling that indicates intolerance of loss of function mutations (Ref: Lek et al., 2016, Nature Genetics). When we used these newer data we observe a significant enrichment for haploinsufficient genes in the Class P (present in pathogenic CNVs) genes compared to Class B. Additionally, we obtained a recent set of genes involved in protein complexes from The Uniprot Consortium. Similar to above we observe a significant enrichment in the Class P genes compared to Class B. We have updated this section to reflect these new results.

5. pg7. 390 CNVRs while on pg 5 they are 167 CNVRs. Please make clear when the peaks are used and when the total regions are used. For pathogenicity I suggest to use only the peak regions

Response: the figures on page 7 (390 CNVRs) refer to duplication and deletion CNVRs separately, whereas on page 5 the 167 CNVRs have grouped deletions and duplications into combined CNVRs when they overlapped. As the mechanistic basis for dosage sensitivity might differ from duplication to deletion CNVs we decided to consider them separately in the section on page 7 where we are trying to isolate candidate genes. However, we see that the text does not make this clear and we have added explanation and justification.

6. pg7. Regarding gene expression levels, I suggest to use the GTEX public data. Genes have different levels of expression in different organs and these could range for more than a log₁₀ per gene in the different organs. I suggest to take the tissue with the highest RPKMs.

Response: we have updated this analysis to use the GTEX data and the conclusions are unchanged.

7. Is there any difference of old versus young genes ?

Response: With this comment we think that the reviewer is referring to the fact that we have excluded "newer genes" from the analysis of conservation across the mammalian genomes. This was a very deliberate step because it is not trivial to distinguish a genuinely new lineage-specific gene from a fast-evolving

gene which is not detectable in older lineages but for different reasons (e.g. see McLysaght & Hurst Nat Rev Genet 2016, <http://www.nature.com/nrg/journal/v17/n9/abs/nrg.2016.78.html>). As we are very specifically interested in changes of copy number, the latter situation would give a false impression of gene absence, and thus would confound the analysis. We have therefore not compared genes of different apparent ages and instead imposed a criterion that all of the genes must be inferred to be present in the common ancestor of mammals (ensembl gene family size >0 at mammalian root) to avoid inference problems. Were we to attempt to compare genes according to age, we predict that the analysis would be hampered by uncertainties in the gene dating.

However, we do note in the Discussion section that these genes that are not inferred to be present in the outgroup (i.e., likely to be either young genes or quickly evolving genes) are enriched in benign CNVs, which is consistent with them being under lower constraint – an expectation true of both young genes and genes with less sequence constraint (faster evolving).

8. pg10. More info on the mouse CNVs is needed. Which regions? From which ref? Is there any pathogenicity attached to the mouse CNVs?

Response: Mouse CNVs were obtained from Pezer et al. 2015 (10.1101/gr.187187.114). No determined pathogenicity is attached to these data as they were identified in wild-caught individuals from four populations and so are presumed to be healthy control variation. CNVs are distributed genome-wide with identified CNVs present on all mouse chromosomes, including both sex chromosomes. This information has been added to the methods section.

9. Fig3a. Is there any pvalue attached to each of the pathogenic duplications/deletions ? What is the null expectation for each one?

Response: The null hypothesis is that there is no difference in evolutionary conservation inside and outside pathogenic CNV regions. We compared the numbers of completely conserved genes (“unchanged”) inside the CNV critical regions with the remainder of the CNVR and the flanking regions. A Mann Whitney U test rejects this null hypothesis (Bonferroni-corrected $P=0.0028$). There is no significant difference between the medians of total pathogenic CNV and the flanking regions.

Reviewer #2 (Remarks to the Author):

This study examines the nature of pathogenic and benign copy number variants in humans. The findings reveal that those that are pathogenic contain genes that are dosage sensitive, involved with various developmental processes and evolutionarily constrained while benign cases are not similarly enriched for these classes of genes. The paper is well written and clear; the results seem reasonable and the arguments sound.

Some minor typos/clarifications:

In the section entitled "Pathogenic CNV are enriched for developmental genes", in the second line a semicolon should follow the citation brackets and a comma should follow "however".

Done

In the second to last paragraph of the Discussion, it is stated that haploinsufficient genes are logically distinct from dosage balanced genes but notes that they are often conflated. It seems that haploinsufficient and dosage balanced genes could in fact overlap (rather than being "conflated") because they might both produce a phenotype from a heterozygous knockout. Semantics, yes, but maybe this could be rephrased?

Done

Reviewer #3 (Remarks to the Author):

Rice and McLysaght present a study of genes found within pathogenic CNVs and their evolutionary constraints based on those submitted in dbVar. This is in general an interesting topic, as, despite many CNVs are reported as being pathogenic, the underlying gene(s) causing the phenotype is often still unknown.

Identifying key features of genes within these CNVs could contribute the identification of specific disease genes. After reading this manuscript I have a few suggestions.

Major:

1. As the authors mention the vast majority of pathogenic CNVs (submitted to dbVar) have till now been identified in patients with a neurodevelopmental disorder; largely due to current diagnostic strategies. This is in contrast to the knowledge that pathogenic CNVs cause a much broader range of phenotypes (for example, but not limited to, Glessner 2014, Orange 2011, Shearer 2014).

Response: these references have been added

This bias in the input dataset, likely to impact the results presented here, in particular the finding that the pathogenic CNVs are enriched in developmental genes, as the majority of the patients are affected by a developmental disorder. Either the conclusions from this result need to be reassessed ie. that pathogenic CNVs in developmental disorders are enriched for developmental genes, or the impact of the bias on the list of CNVs determined.

Response: We have added the following text to the end of the discussion:

“There is great interest in the relationship between development and dosage sensitivity and CNVs in general. As we noted, there is a potential bias in the annotation of disease CNVs due to this interest and due to the fact that developmental disorders are expected to be more reliably identified at the time of sample collection. Therefore, the enrichment for developmental genes in pathogenic CNVs must normally be interpreted in that light. However, the evolutionary measures based on conservation of copy number across mammalian species are independent of disease annotation and have no such reporter or study bias. Our finding that these evolutionarily constrained genes are indeed enriched for developmental genes confirms the view of development as an inherently dosage sensitive process.”

2. The authors conclude that genes found with in pathogenic CNVs are evolutionary conserved in copy number, these findings should be discussed within context of previous results on evolutionary forces within CNVs, for example Nguyen et al Genome Research 2008 and Huang et al Plos Genetics 2010.

Response: We now cite both of these papers as previously having shown more duplications of genes in benign CNVRs and haplosufficient genes respectively.

These are mentioned both within the results section and also in the Discussion.

3. p6 "Clustering of developmental genes may contribute to their pathogenicity". This concept will influence the results presented here. More analysis of this concept independent to the pathogenic CNVs is required.

Response: The last paragraph of the section "Pathogenic genes are enriched for CNVs" describes a test for whether the clustering of developmental genes is driving the observation that pathogenic peak regions are enriched for developmental genes, and we show that it is not.

To investigate if clustering of developmental genes influences our evolutionary analysis, we looked for a correlation between distance between developmental genes and the number of mammalian genomes where ortholog copy number is unchanged (where one-to-one orthologous relationships exist). If clustering has an effect, we would expect greater conservation of clustered developmental genes than of isolated developmental genes. To test this, developmental genes were grouped according to the number of genomes with unchanged copy number and for each group a distribution of distances to the nearest developmental gene was plotted (Supp Figure 7). This test is completely independent of the clinical classification of CNVs. We observe no effect of the distance to nearest developmental gene on the evolutionary conservation. This is now mentioned in the section "Genes with conserved copy number in mammals are enriched for ...".

Minor:

1. p12 Methods, more insights should be given on the datasets included in the study, it is unclear the range of data which has been included, patient numbers, study cohorts...

Summary data on CNVs included from specific studies are now listed in Supp Table 1.

2. Table 1, requires more explanation in terms of footnotes, for example CNL, CNG

Done.

3. More details are required of how "developmental genes" are defined, it is currently unclear how this set has been defined.

Developmental genes were defined as those with GO term "developmental

process (GO:0032502)". This is now explicit in the methods.

4. Table 2, abbreviations should be more clearly explained. For example BL/PG

We have added footnotes to this effect.

5. p5 final paragraph "Although the lengths of full benign CNV...are more similar"; this should be quantified.

These numbers have been added to the text

6. Figure 2d, the sample size (n) should be clearly stated.

This information has been added underneath each boxplot column

Reviewers' Comments:

Reviewer #1 (Remarks to the Author)

Editorial Note: Comments were provided to the Editor only. The reviewer had no further concerns with the manuscript.

Reviewer #3 (Remarks to the Author)

The authors have sufficiently addressed my comments.